# Multi-Morbidity and Risk of Breast Cancer among Women in the UK Biobank Cohort

**DOI:** 10.3390/cancers15041165

**Published:** 2023-02-11

**Authors:** Afi Mawulawoe Sylvie Henyoh, Rodrigue S. Allodji, Florent de Vathaire, Marie-Christine Boutron-Ruault, Neige M. Y. Journy, Thi-Van-Trinh Tran

**Affiliations:** 1Radiation Epidemiology Group, Center for Research in Epidemiology and Population Health, INSERM U1018, Paris Sud-Paris Saclay University, Gustave Roussy, 94800 Villejuif, France; 2Health across Generations Team, Center for Research in Epidemiology and Population Health, INSERM U1018, Paris Sud-Paris Saclay University, Gustave Roussy, 94800 Villejuif, France

**Keywords:** morbidity, morbidity patterns, breast cancer, incidence, cohort study, multiple correspondence analysis, cluster analysis

## Abstract

**Simple Summary:**

(Multi-)Morbidity shares common biological mechanisms or risk factors with breast cancer. However, the risk of breast cancer among women with (multi-)morbidity remains unclear. In this study, using data of 239,436 female participants aged 40–69 of the UK Biobank cohort, we identified five chronic disease patterns: no-predominant morbidity, psychiatric morbidities, respiratory/immunological morbidities, cardiovascular/metabolic morbidities, and unspecific morbidities. After a median follow-up of 7 years, 5326 women developed breast cancer. We found no association between breast cancer risk and either the number of chronic diseases or chronic disease patterns, apart from an increased risk among women aged younger than 50 with a psychiatric pattern. Women with any multi-morbidity were more likely to die or to be diagnosed with other cancers. Our findings suggest that multi-morbidity may not be a key factor to help identify patients at an increased risk of breast cancer.

**Abstract:**

(Multi-)Morbidity shares common biological mechanisms or risk factors with breast cancer. This study aimed to investigate the association between the number of morbidities and patterns of morbidity and the risk of female breast cancer. Among 239,436 women (40–69 years) enrolled in the UK Biobank cohort who had no cancer history at baseline, we identified 35 self-reported chronic diseases at baseline. We assigned individuals into morbidity patterns using agglomerative hierarchical clustering analysis. We fitted Cox models to estimate hazard ratios (HRs) and 95% confidence intervals (CIs) for breast cancer risk. In total, 58.4% of women had at least one morbidity, and the prevalence of multi-morbidity was 25.8%. During a median 7-year follow-up, there was no association between breast cancer risk (5326 cases) and either the number of morbidities or the identified clinically relevant morbidity patterns: no-predominant morbidity (reference), psychiatric morbidities (HR = 1.04, 95%CI 0.94–1.16), respiratory/immunological morbidities (HR = 0.98, 95%CI 0.90–1.07), cardiovascular/metabolic morbidities (HR = 0.93, 95%CI 0.81–1.06), and unspecific morbidities (HR = 0.98, 95%CI 0.89–1.07), overall. Among women younger than 50 years of age only, however, there was a significant association with psychiatric morbidity patterns compared to the no-predominant morbidity pattern (HR = 1.25, 95%CI 1.02–1.52). The other associations did not vary when stratifying by age at baseline and adherence to mammography recommendations. In conclusion, multi-morbidity was not a key factor to help identify patients at an increased risk of breast cancer.

## 1. Introduction

Breast cancer is the most common female cancer, with 2,088,849 new cases worldwide in 2018, accounting for 11.6% of incident cancer cases [1]. Despite decades of intensive research effort, only about 70% of the disease occurrence is explained by well-established risk factors [2]. Most of the identified risk factors are not readily modifiable [2,3,4], leading to a need for additional research to better understand etiologic processes.

In developed countries, most breast cancer cases are diagnosed among women of middle age or older [5], coinciding with the occurrence of other long-term morbidities [6,7]. Previous studies have suggested associations between breast cancer risk and specific chronic diseases, such as endocrine disorders [8,9], inflammatory conditions [10], autoimmune diseases [11], and cardiovascular diseases (CVDs) [12], especially among postmenopausal women. The underlying mechanisms of these associations could involve shared common physiopathological pathways (e.g., estrogen-related pathways, inflammation pathways) [13,14,15,16], shared genetic predispositions, shared risk factors (e.g., obesity, physical inactivity) [17], and medications (e.g., aspirin) [18].

As people get older, they often develop two or more chronic diseases. With an aging population, the number of people experiencing several multi-morbidities is rising globally [6,7,19,20,21]. In the general population, co-existing morbidities could be classified into common clinically meaningful patterns [22,23]. Sharing underlying biological mechanisms and/or sets of risk factors, the morbidities in the same cluster often interact mutually, which complicates treatments and management and increases the risk of adverse events above and beyond the sum of the risk of individual disease [24]. Being diagnosed with multi-morbidity is also associated with an increased likelihood of being subjected to breast cancer screening [25,26,27], which may lead to increased surveillance of breast cancer incidence. Thus, it is necessary to consider patterns of morbidity, in addition to associations with single chronic diseases, with breast cancer risk.

However, to date, there is no epidemiological evidence as to whether and to what extent breast cancer risk varies according to different patterns of morbidity. In this context, our study aimed to investigate the association between the number of morbidities and patterns of morbidity and the risk of female breast cancer.

## 2. Materials and Methods

### 2.1. Data Source and Study Design

The UK Biobank is a prospective population-based cohort that recruited 273,375 women, aged 40 to 69 years, from March 2006 to July 2010 [28]. Individuals were invited to participate on a voluntary basis and provided electronic informed consent for data provision and linkage. The baseline data assessment included self-reported data on personal and family medical history, lifestyle, hormone-related factors, and sociodemographic characteristics. Additional anthropometric measurements were performed. The cohort additionally retrieved individual information from the national cancer and death registries.

### 2.2. Study Population

We excluded women with any cancer diagnosis prior to baseline except non-melanoma skin cancer (n = 29,332), women who underwent a mastectomy prior to baseline (n = 2457), and women with less than one year of follow-up (n = 2150), leaving 239,436 women in the final analysis (Figure 1).

### 2.3. Baseline Morbidity Identification

Based on an established list of morbidities, which was originally designed by Barnett et al. [19] to measure multi-morbidity in a large population-based dataset and subsequently validated in the UK Biobank cohort (Appendix B, Table A1) [29], we defined 35 morbidities based on baseline self-reported health conditions (Figure 2). For each woman, we computed the total number of morbidities and categorized them as none/one/multi-morbidity (at least two morbidities).

### 2.4. Breast Cancer Ascertainment

We defined breast cancer as a diagnosis of invasive or in situ breast cancer, using the international classification of diseases (ICD) versions 9 and 10 (ICD-10: C50 or D05; ICD-9: 174 or 2330). We considered only breast cancer cases that were the first cancer diagnosed.

### 2.5. Baseline Confounding Factors

All confounding factors (age at menarche, age at menopause, menopausal hormone therapy use, oral contraceptive use, parity and age at first birth, body mass index (BMI), ethnicity, Townsend score, level of physical activity, alcohol consumption) were measured/collected at baseline. We selected well-established breast cancer risk factors based on previous studies [30,31]. We also selected variables that were statistically significantly associated with both morbidity and breast cancer risk (*p*-value < 0.05) as confounders if their inclusion in the age-adjusted Cox models changed the hazard ratio by 5% or more [32]. See Appendix B, Table A2 for more details on the variables of interest, their definition, and information sources.

### 2.6. Statistical Analysis

#### 2.6.1. Multiple Correspondence Analysis (MCA) and Cluster Analysis (See Appendix C)

Among 35 baseline self-report morbidities, we included only morbidities with a prevalence of more than 1% (Figure 2) to obtain stable clustering results [33]. We used MCA [34,35] and cluster analysis to identify morbidity patterns. MCA can produce the input data for the cluster analysis, while reducing noise by excluding unnecessary dimensions that do not contribute significantly to the cluster’s classification. We determined the optimal number of dimensions to extract based on the elbow rule in the Scree plot [34] and Horn’s parallel analysis for common factor analysis [36].

Using the numerical outputs of the MCA, we performed agglomerative hierarchical clustering (AHC) preceded by K-means clustering with 2000 initial cluster seeds [37], through the HCPC function of the Factominer package in R. This method allowed us to reduce the required memory allocations [38,39]. We considered the distance between points in Euclidean space as the distance metric [40], and Ward’s method was used to create homogeneous clusters by fusion [36]. We chose the optimal number of clusters, i.e., the identified morbidity patterns and assessed cluster quality, using the Davies–Bouldin [41] and the GAP indexes [42]. The optimal number of clusters was the one that corresponded to the minimum value of the Davies–Bouldin index and to the maximum Gap statistics index.

Within each cluster, we computed the observed/expected ratios (“O/E-ratios”) for each single morbidity, i.e., the ratio between the prevalence of a given condition in a cluster and its prevalence in the overall study population. Similarly, we computed the exclusivity of each single morbidity, i.e., the number of individuals that had a given morbidity in a cluster over the number of individuals with the same morbidity in the whole study population. A morbidity was considered part of a given morbidity cluster when its O/E-ratio was ≥2 and its exclusivity was ≥25% [23,43]. We named the morbidity patterns based on the predominant morbidities in the clusters.

#### 2.6.2. Association among the Number of Morbidities, Morbidity Patterns, and Breast Cancer Risk

The follow-up time started at the date of first registration at a UK Biobank center and ended at the date of the first cancer diagnosis (any cancer diagnosis, except non-melanoma skin cancer) or mastectomy, death, loss to follow-up, or 31 March 2016, whichever came first. We fitted Cox proportional hazard models to estimate hazard ratios and 95% confidence intervals (95%CIs) of breast cancer risk associated with each single pre-existing baseline morbidity included in the cluster analysis, the number of morbidities, and the morbidity patterns. The timescale was the follow-up time.

We graphically assessed the proportional hazards assumption using scaled Schoenfeld residuals plots and log linearity assumption (for quantitative covariates) using Martingale residuals plots and deviance residuals plots. The final multivariable Cox models were adjusted for age at baseline, age at menarche, age at menopause, menopausal hormone therapy use, oral contraception use, parity and age at first birth, BMI, ethnicity, the Townsend score, level of physical activity, and alcohol consumption.

We tested the modifying effects of age at baseline, the adherence to the recommendations for breast cancer screening, the BMI, the socioeconomic status, the physical activity level, and the menopause status at baseline with the likelihood ratio test. We conducted several sensitivity analyses: (i) we restricted analyses to menopausal women; (ii) we considered only invasive breast cancer as the outcome; (iii) we used the attained age as the timescale; (iv) we considered death and diagnosis of non-breast cancer as competing risks, using sub-distribution hazards models [44]; (v) we extracted 11 MCA dimensions, which accounted for more than 70% of the total variability among the study population, as recommended by Higgs [45]; we also extracted all dimensions, assuming they were all significant, and kept different numbers of clusters (3 and 4 clusters) with both 11 and all dimensions extracted.

All statistical analyses were performed using R version 4.1.0.

## 3. Results

In the study population, the median age at baseline was 57.7 years (interquartile range [IQR]: 50.2, 63.2). At least one morbidity was present in 58.4% of women at baseline, and the prevalence of multi-morbidity was 25.8%. Hypertension was the most prevalent morbidity (23.1%), followed by painful conditions (17.2%) and asthma (12.3%). The prevalence of obesity was 23.5%, and 23.5% of women had menopause after the age of 51 at baseline. Most women were postmenopausal (73.5%) and were adherent to breast cancer screening recommendations (66.6%) at baseline, as assessed at recruitment (Table 1, Figure 2). During a median follow-up time of 7.1 years (IQR: 6.4, 7.8), 5,326 women developed breast cancer (2.0%).

### 3.1. Description of Morbidity Patterns 

We considered the first five MCA dimensions (see Appendix A), which explained 39% of the total variance, as input to the clustering algorithms. We identified five baseline morbidity patterns (see Appendix A), named as follows: Pattern 1—no-predominant morbidity, pattern 2—psychiatric morbidities, pattern 3—respiratory/immunological morbidities, pattern 4—cardiovascular/metabolic morbidities, pattern 5—unspecific morbidities (see Table 1).

#### 3.1.1. Pattern 1: No-Predominant Morbidity [n = 159,083 (66.4%), 3534 Breast Cancer Cases (2.0% of Cases)]

The median age at baseline was 57.4 years (IQR: 49.9, 63.0), and the median follow-up time was 7.1 years (IQR: 6.4, 7.8). There was no morbidity with an O/E ratio ≥ 2. The main features of this pattern were the low rate of multi-morbidity (6.9%) and the high rate of the absence of morbidity (62.6%).

#### 3.1.2. Pattern 2: Psychiatric Morbidities [n = 16,627 (7.0%), 381 Breast Cancer Cases (2.0% of Cases)]

The median age at baseline was 55.7 years (IQR: 48.7, 61.7), and the median follow-up time was 7.0 years (IQR: 6.3, 7.8). Women with this pattern were predominantly diagnosed with anxiety and depression disorders.

#### 3.1.3. Pattern 3: Respiratory/Immunological Morbidities [n = 27,920 (11.7%), 611 Breast Cancer Cases (2.0% of cases)]

The median age at baseline was 56.7 years (IQR: 49.1, 62.8), and the median follow-up time was 7.1 years (IQR: 6.4, 7.8). Women with this pattern were predominantly diagnosed with psoriasis/eczema, COPD, and asthma.

#### 3.1.4. Pattern 4: Cardiovascular/Metabolic Morbidities [n = 11,041 (4.6%), 246 Breast Cancer Cases (2.0% of cases)]

The median age at baseline was 62.6 years (IQR: 57.2, 66.4), and the median follow-up time was 7.0 years (IQR: 6.3, 7.8). Women with this pattern were predominantly diagnosed with diabetes, stroke, and coronary–heart disease. The main features of this pattern were the high proportions of elderly (about 65% were 65 years or older at baseline), multi-morbidity (96.6%), and deprived people (37.1% of women with this pattern were in the quintile with the highest levels of deprivation).

#### 3.1.5. Pattern 5: Unspecific Morbidities [n = 24,765 (10.3%), 554 Breast Cancer Cases (2.0%)]

The median age at baseline was 59.2 years (IQR: 51.9, 64.0), and the median follow-up time was 7.1 years (IQR: 6.4, 7.8). Women with this pattern were predominantly diagnosed with migraine, diverticular intestine disease, inflammatory bowel disease, rheumatoid disease, and threated dyspepsia.

### 3.2. Breast Cancer Risk According to the Number of Morbidities and Morbidity Patterns

In both age-adjusted and fully adjusted models, no significant association was found between either the number of morbidities or any morbidity pattern and breast cancer risk, but there was a 12% increased risk associated with self-reported depression (Table 2 and Table 3). The results did not vary significantly with age at baseline (*p*-value interaction = 0.43 and 0.07, for the analyses on the number of morbidities and morbidity patterns, respectively) and adherence to recommendations for breast cancer screening among women aged 50 and older (*p*-value interaction = 0.44 and 0.84, for the analyses on the number of morbidities and morbidity patterns, respectively), although we found an increased risk among women aged of up to 50 years in the psychiatric morbidities pattern (HR= 1.25; 95%CI: 1.02–1.52) (Figure 3 and Figure 4). The results remained consistent after accounting for competing risks (Table 4), when considering attained age as the timescale in the Cox models (Appendix A) and in other sensitivity analyses (see Appendix A).

## 4. Discussion

Among female participants in the UK Biobank cohort, 58.4% had at least one chronic disease, while 25.8% had two or more simultaneous morbidities. Hypertension was the most prevalent disease (23.1%) at baseline. We found five morbidity patterns: no-predominant morbidity, psychiatric morbidities, respiratory/immunological morbidities, cardiovascular/metabolic morbidities, and unspecific morbidities. There was a 1.12-fold increased risk among women who self-reported depression and a 25% increased risk of breast cancer associated with a psychiatric morbidity pattern compared to that with the no-predominant morbidity pattern, among women younger than 50 only. We did not observe other significant associations between either the number of morbidities or any morbidity pattern and the risk of breast cancer, which did not vary according to adherence to breast cancer screening recommendations, socioeconomic status, BMI, physical activity level, or menopausal status.

Despite heterogeneous findings in previous studies on morbidities across different populations and settings, several morbidity patterns often emerge in the literature, which were also observed in our study [22,23,46,47]. The pattern of cardiovascular/metabolic morbidities has been extensively described previously, as there are established etiologic associations among diabetes, stroke, heart failure, and heart disease, with an interlinked pathophysiology and common risk factors, such as obesity, physical inactivity, and smoking [48]. For the pattern of psychiatric morbidities, although little is known about the pathogenesis of depression and anxiety, these two frequent mental illnesses share a largely overlapping set of risk factors with breast cancer, including female sex, genetic predisposition, family history, and environmental influence (childhood adversity, low socioeconomic status) [49,50]. Depression and anxiety are also common coexisting conditions among patients with chronic comorbidities, including cancer [51,52]. Consistent with our findings, a recent nationwide population-based study has shown that mental disorders were associated with a subsequent higher risk of cancer, although the causal link remains a topic of debate [52]. The diseases included in the respiratory pattern, such as chronic obstructive pulmonary disease and asthma, involve a prolonged inflammatory response and the sharing of risk factors, such as smoking, an unhealthy diet, physical inactivity, and high alcohol consumption. However, combinations among asthma, COPD, and psoriasis and eczema are less common. Thus, these patterns found in our clustering analysis not only represent a clinically relevant morbidity status in women in the UK Biobank cohort but also reflect distinct profiles of (known or unknown), shared genetics, and behavioral and environmental risk factors, both of which might increase the risk of developing cancer.

Indeed, to our knowledge, our study is the first to investigate the association between morbidity patterns and breast cancer risk. We found no association between either the number of morbidities or morbidity patterns and breast cancer risk, regardless of the women’s age at baseline, and socioeconomic characteristics, apart from an increased risk among women aged less than 50 having multiple psychiatric diseases. Analyses stratified based on adherence to breast cancer screening recommendations did not modify our main results, suggesting that surveillance bias is not an important modifying factor in the association between breast cancer risk and morbidities. Previously, there was only a case-control study reporting results on the association between multi-morbidity and breast cancer risk. The findings indicated that an increasing number of morbidities measured with the Charlson comorbidity index (CCI) was associated with an increasing breast cancer risk (46,324 cases) after a 10-year follow-up of women aged 45–85, but no association was found for individual morbidities [53]. However, they were not able to control for confounding factors other than age at baseline and to account for surveillance bias. For comparison purposes, we applied the same methods in an additional analysis by using the CCI (Appendix A), and we did not find a significant association between the Charlson morbidity number and breast cancer risk after adjusting for well-known risk factors.

There are several hypotheses to explain the null results. First, women with morbidity could experience other serious long-term outcomes before a breast cancer diagnosis. Indeed, when accounting for death and malignancies other than breast cancer as competing risks, we found that compared to that in women with no predominant morbidity, women with other patterns were more likely to die and/or to be diagnosed with other cancers. This is particularly pronounced among women with cardiovascular/metabolic and respiratory/immunological morbidities. Second, given the different biologic characteristics of divergent breast cancer subtypes [54] and the complexity of multi-morbidity mechanisms and risk factors, the risk estimations could vary across individual associations, and the possible opposing effects could drive the combined estimates toward null. For instance, BMI, a common risk factor of various morbidities, is strongly associated with hormone receptor-positive tumors, but not a triple-negative or core basal phenotype [55]. A high BMI is a risk factor of postmenopausal breast cancer, but a protective factor of premenopausal breast cancer. Type 2 diabetes is an independent risk factor of breast cancer risk in postmenopausal women, but no increased risk was observed for premenopausal women [8]. In our study, when restricting analyses to postmenopausal women only, the null associations remained consistent. Previous large prospective cohorts reported that low socioeconomic positions, a contributing factor of psychiatric morbidities, were found to be associated with a lower risk of ER+ breast cancer but a higher risk of the ER- subtype [56,57]. Meanwhile, adverse life events, such as childhood abuse and divorce, were associated with a higher risk of ER+, but not ER-, breast cancer [57,58]. Third, our null results could also suggest that the underlying common biological pathways among morbidities in an individual pattern and their shared risk factors were not a key factor explaining breast cancer risk after accounting for established breast cancer risk factors.

Strength and limitations: The UK Biobank cohort is a large population-based cohort with a high follow-up rate and important number of breast cancer cases. The cohort includes a wide range of information on personal medical history, reproductive factors, lifestyle factors, socioeconomic status, and family medical history, with low levels of missing data. Nevertheless, there are several limitations that must be acknowledged. Assuming that the prevalence of having at least one morbidity in women in the UK Biobank cohort is slightly lower than what has been found (42.2%, 33.8%) in previous studies of Barnett and Gondek, respectively (since these studies have included data of both women and men in the analyses, which could lead to a potential underestimation of the morbidity prevalence), this suggests the occurrence of “healthy” volunteer bias (i.e., UK Biobank participants are more likely to be in good health conditions than the general population) [59,60]. However, since our study focuses on investigating breast cancer risk in relation to morbidity and not on estimating disease prevalence rates and many people with a wide range of morbidities and risk factors are included in the cohort, the risk estimations are unlikely to be biased [59,60]. We used self-reported health condition data, which were not externally validated, and the UK Biobank did not include information on morbidity severity. There was no longitudinal updated morbidity status and thus no possibility to study changes in morbidity patterns during follow-up. We also missed details on the breast cancer stage, grade, and receptor status. This did not allow us to further study the surveillance biases related to the disease stage and grade or to investigate potential pathways related to tumor receptor status.

## 5. Conclusions

Female participants in the UK Biobank cohort can be classified into five morbidity patterns: no-predominant morbidity, psychiatric morbidities, respiratory/immunological morbidities, cardiovascular/metabolic morbidities, and unspecific morbidities. We found a significant increased risk among women aged younger than 50 with a psychiatric diseases pattern, but there was no other significant association among the number of morbidities, the morbidity patterns, and the risk of breast cancer in this population. Our findings suggest that multimorbidity is not a decisive factor to help identify patients at increased risk of breast cancer.

## Figures and Tables

**Figure 1 cancers-15-01165-f001:**
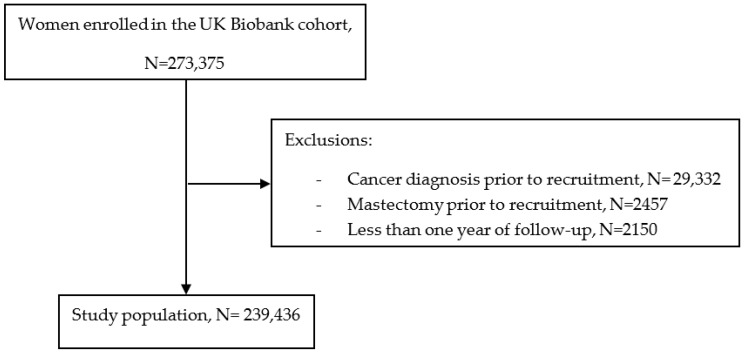
Flow chart of the study population.

**Figure 2 cancers-15-01165-f002:**
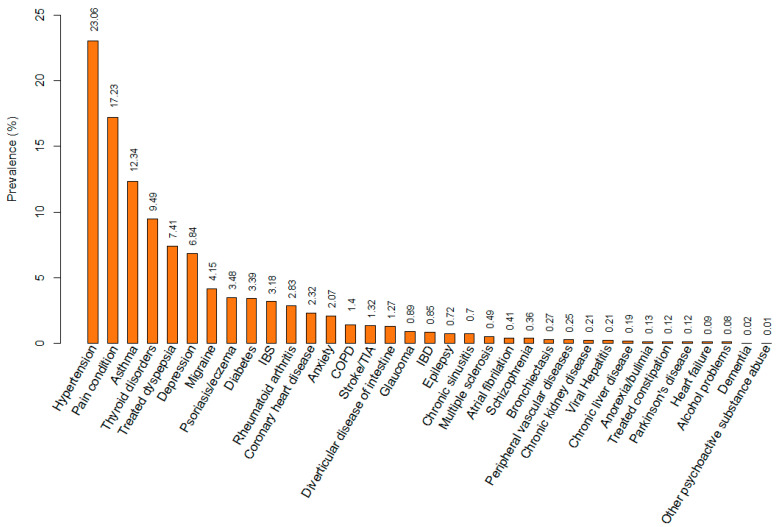
Morbidities identified among cancer-free UK Biobank women at recruitment.

**Figure 3 cancers-15-01165-f003:**
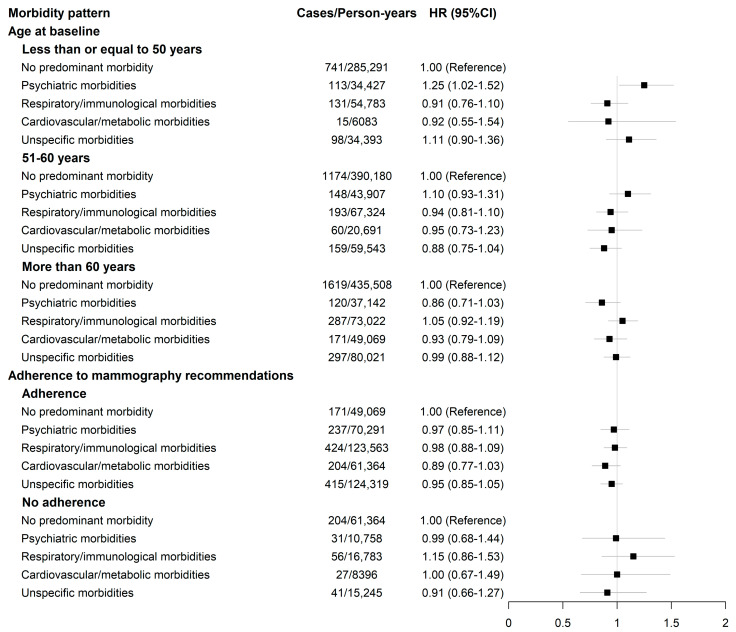
Associations between morbidity clusters and breast cancer risk, according to age-groups and the adherence to breast cancer screening recommendations. HR: hazard ratio; CI: confidence interval. The adherence to mammography included only women older than 50 years. The model was adjusted for age at menarche (continuous), age at menopause (still had periods; had menopause before the age of 45 years; had menopause between the age of 45 and 54; had menopause after the age of 55), menopausal hormone therapy use (never; ever, less than 5-year duration; ever, 5 years and longer; ever, unknown duration), oral contraceptive use (never; ever, less than 10-year duration; ever, at least 10-year duration; ever, unknown duration; unknown status), parity and age at first birth (no live birth; at least one birth before age 30; at least one birth after age 30), BMI (continuous), ethnicity (Asian; Black/Caribbean; White; others/unknown), Townsend score (continuous); level of physical activity (low; moderate; high), alcohol consumption (never; twice a week or less; three times a week or more; unknown status).

**Figure 4 cancers-15-01165-f004:**
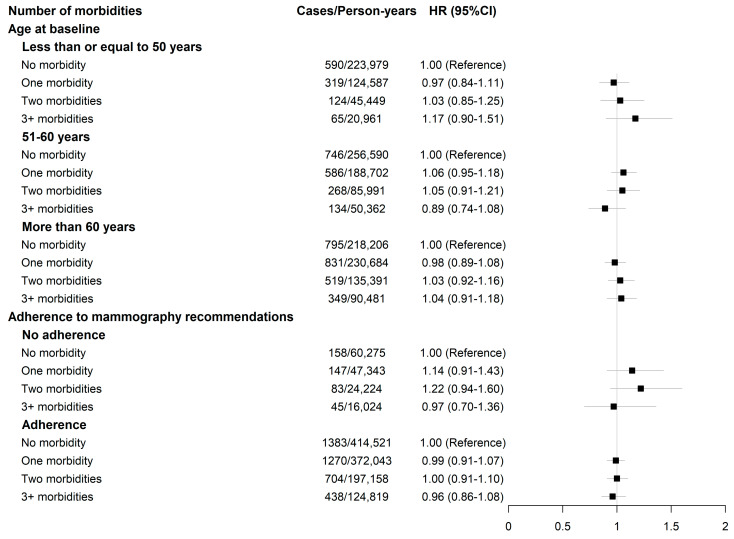
Associations between morbidity clusters and breast cancer risk, according to age groups and the adherence to breast cancer screening recommendations. HR: hazard ratio; CI: confidence interval. The adherence to mammography included only women older than 50 years. The model was adjusted for age at menarche (continuous), age at menopause (still had periods; had menopause before the age of 45 years; had menopause between the age of 45 and 54; had menopause after the age of 55), menopausal hormone therapy use (never; ever, less than 5-year duration; ever, 5 years and longer; ever, unknown duration), oral contraceptive use (never; ever, less than 10-year duration; ever, at least 10-year duration; ever, unknown duration; unknown status), parity and age at first birth (no live birth; at least one birth before age 30; at least one birth after age 30), BMI (continuous), ethnicity (Asian; Black/Caribbean; White; others/unknown), Townsend score (continuous); level of physical activity (low; moderate; high), alcohol consumption (never; twice a week or less; three times a week or more; unknown status).

**Table 1 cancers-15-01165-t001:** Characteristics of the overall study population and the identified baseline morbidity patterns.

**Characteristics**	**Overall Study Population** **N = 239,436**	**Pattern 1: No-Predominant Morbidity** **N = 159,083**	**Pattern 2: Psychiatric Morbidities** **N = 16,627**	**Pattern 3: Respiratory/Immunological Morbidities** **N = 27,920**	**Pattern 4:** **Cardiovascular/Metabolic Morbidities** **N = 11,041**	**Pattern 5: Unspecific Morbidities** **N = 24,765**	***p*-Value ***
**Year of follow-up, median (IQR)**	7.1 (6.4, 7.8)	7.1 (6.4, 7.8)	7.0 (6.3, 7.8)	7.1 (6.4, 7.8)	7.0 (6.3, 7.8)	7.1 (6.4, 7.8)	<0.001
**Breast cancer cases, n (%)**	5326 (2)	3534 (2)	381 (2)	611 (2)	246 (2)	554 (2)	0.97
**Number of comorbid conditions, n (%)**							<0.001
None	99,614 (41.6)	99,614 (62.6)	0 (0.0)	0 (0.0)	0 (0.0)	0 (0.0)	
One	77,994 (32.6)	48,489 (30.5)	6260 (37.6)	14,283 (51.2)	379 (3.4)	8583 (34.7)	
Two	38,424 (16.0)	10,145 (6.4)	5974 (35.9)	9156 (32.8)	4717 (42.7)	8432 (34.0)	
Three and more	23,404 (9.8)	835 (0.5)	4393 (26.4)	4481 (16.0)	5945 (53.8)	7750 (31.3)	
**Morbidity, n (%)**							
Stroke and transient ischemic attack (TIA)	3149 (1.3)	833 (0.5)	90 (0.5)	181 (0.6)	1761 (15.9)	284 (1.1)	<0.001
Diabetes	8122 (3.4)	1429 (0.9)	63 (0.4)	282 (1.0)	5924 (53.7)	424 (1.7)	<0.001
Coronary heart disease	5566 (2.3)	796 (0.5)	53 (0.3)	329 (1.2)	3978 (36.0)	410 (1.7)	<0.001
Migraine	9947 (4.2)	247 (0.2)	940 (5.7)	686 (2.5)	102 (0.9)	7972 (32.2)	<0.001
Diverticular disease of intestine	3048 (1.3)	0 (0.0)	10 (0.1)	109 (0.4)	247 (2.2)	2682 (10.8)	<0.001
Irritable bowel syndrome	7622 (3.2)	32 (0.0)	642 (3.9)	195 (0.7)	276 (2.5)	6477 (26.2)	<0.001
Rheumatoid arthritis	6778 (2.8)	0 (0.0)	81 (0.5)	201 (0.7)	4 (0.0)	6492 (26.2)	<0.001
Treated dyspepsia	17,733 (7.4)	6427 (4.0)	1704 (10.2)	2053 (7.4)	1807 (16.4)	5742 (23.2)	<0.001
Psoriasis or eczema	8344 (3.5)	0 (0.0)	773 (4.6)	5823 (20.9)	190 (1.7)	1558 (6.3)	<0.001
Chronic obstructive respiratory disease (COPD)	3355 (1.4)	0 (0.0)	7 (0.0)	3333 (11.9)	2 (0.0)	13 (0.1)	<0.001
Asthma	29,541 (12.3)	0 (0.0)	2311 (13.9)	21,708 (77.8)	2473 (22.4)	3049 (12.3)	<0.001
Anxiety	4964 (2.1)	0 (0.0)	4460 (26.8)	113 (0.4)	216 (2.0)	175 (0.7)	<0.001
Depression	16,368 (6.8)	0 (0.0)	13,362 (80.4)	424 (1.5)	1157 (10.5)	1425 (5.8)	<0.001
Thyroid disorders	22,718 (9.5)	13,277 (8.3)	1776 (10.7)	2213 (7.9)	2806 (25.4)	2646 (10.7)	<0.001
Hypertension	55,223 (23.1)	31,013 (19.5)	3647 (21.9)	6112 (21.9)	8505 (77.0)	5946 (24.0)	<0.001
Pain conditions	41,258 (17.2)	21,363 (13.4)	3665 (22.0)	4767 (17.1)	3132 (28.4)	8331 (33.6)	<0.001
**Age at baseline, median (IQR)**	57.7 (50.2, 63.2)	57.4 (49.9, 63.0)	55.7 (48.7, 61.7)	56.7 (49.1, 62.8)	62.6 (57.2, 66.4)	59.2 (51.9, 64.0)	<0.001
**Family history of breast cancer, n (%)**	25,330 (10.6)	16,858 (10.6)	1765 (10.6)	2885 (10.3)	1102 (10.0)	2720 (11.0)	0.035
BMI, n (%)							<0.001
<18.5	1803 (0.8)	1215 (0.8)	115 (0.7)	225 (0.8)	23 (0.2)	225 (0.9)	
18.5–25	92,857 (38.8)	66,570 (41.8)	5644 (33.9)	10,139 (36.3)	1547 (14.0)	8957 (36.2)	
25–30	87,381 (36.5)	58,431 (36.7)	6067 (36.5)	10,161 (36.4)	3581 (32.4)	9141 (36.9)	
>30	56,150 (23.5)	31,992 (20.1)	4725 (28.4)	7282 (26.1)	5799 (52.5)	6352 (25.6)	
Unknown	1245 (0.5)	875 (0.6)	76 (0.5)	113 (0.4)	91 (0.8)	90 (0.4)	
**Adherence to breast cancer screening, n (%)**							<0.001
<50 years of age	58,722 (24.5)	40,371 (25.4)	4902 (29.5)	7745 (27.7)	873 (7.9)	4831 (19.5)	
>50 years of age, >3 years ago	7929 (3.3)	5072 (3.2)	554 (3.3)	889 (3.2)	545 (4.9)	869 (3.5)	
>50 years of age, in the last 3 years	159,407 (66.6)	104,789 (65.9)	10,158 (61.1)	17,761 (63.6)	8937 (80.9)	17,762 (71.7)	
>50 years of age, never	8013 (3.3)	5384 (3.4)	631 (3.8)	943 (3.4)	348 (3.2)	707 (2.9)	
>50 years of age, unknown	5365 (2.2)	3467 (2.2)	382 (2.3)	582 (2.1)	338 (3.1)	596 (2.4)	
**Age at menarche, median (IQR)**	13.0 (12.0, 14.0)	13 (12.0, 14.0)	13 (12.0, 14.0)	13.0 (12.0, 14.0)	13 (12.0, 14.0)	13 (12.0, 14.0)	<0.001
**Age at menopause ^µ^, median (IQR)**	50.0 (47.0, 52.0)	50.0 (47.0, 52.0)	50.0 (45.5, 52.0)	50.0 (46.0, 52.0)	50.0 (45.0, 52.0)	50.0 (46.0, 52.0)	<0.001
**Menopause status at baseline, n (%)**							<0.001
Still had periods	63,488 (26.5)	44,275 (27.8)	4979 (29.9)	8152 (29.2)	951 (8.6)	5131 (20.7)	
Had menopause before the age of 45	25,659 (10.7)	14,768 (9.3)	2095 (12.6)	3356 (12.0)	2024 (18.3)	3416 (13.8)	
Had menopause between the age of 45 and 54	129,114 (53.9)	85,911 (54.0)	8332 (50.1)	14,084 (50.4)	6796 (61.6)	13,991 (56.5)	
Had menopause after the age of 54	21,175 (8.8)	14,129 (8.9)	1221 (7.3)	2328 (8.3)	1270 (11.5)	2227 (9.1)	
**Menopausal hormone therapy use ^µ^, n (%)**							<0.001
Never	85,613 (48.7)	59,734 (52.0)	4572 (39.3)	8935 (45.2)	4485 (44.4)	7887 (40.2)	
Ever, less than 5 years duration	31,000 (17.6)	19,322 (16.8)	2553 (21.9)	3683 (18.6)	1620 (16.1)	3822 (19.5)	
Ever, 5 years and longer duration	47,233 (26.8)	28,799 (25.1)	3496 (30.0)	5759 (29.1)	2898 (28.7)	6281 (32.0)	
Ever, unknown duration	11,229 (6.4)	6386 (5.6)	975 (8.4)	1314 (6.6)	1004 (9.9)	1550 (7.9)	
Unknown status	874 (0.5)	567 (0.5)	52 (0.4)	77 (0.4)	84 (0.8)	94 (0.5)	
**Oral contraception use, n (%)**							<0.001
Never	44,767 (18.7)	29,175 (18.3)	2795 (16.8)	4818 (17.3)	3147 (28.5)	4832 (19.5)	
Ever, less than 10 years duration	87,270 (36.4)	57,671 (36.3)	5929 (35.7)	10,134 (36.3)	4074 (36.9)	9462 (38.2)	
Ever, 10 years and longer duration	84,462 (35.3)	57,626 (36.2)	6117 (36.8)	10,315 (36.9)	2505 (22.7)	7899 (31.9)	
Ever, unknown duration	22,542 (9.4)	14,354 (9.0)	1758 (10.6)	2628 (9.4)	1270 (11.5)	2532 (10.2)	
Unknown status	395 (0.2)	257 (0.2)	28 (0.2)	25 (0.1)	45 (0.4)	40 (0.2)	
**Parity and age at first birth, n (%)**							<0.001
None of live birth	44,601 (18.6)	29,572 (18.6)	3575 (21.5)	5497 (19.7)	1614 (14.6)	4343 (17.5)	
At least one birth before 30	150,386 (62.8)	98,115 (61.7)	10,088 (60.7)	17,341 (62.1)	8183 (74.1)	16,659 (67.3)	
At least one birth after age 30	43,302 (18.1)	30,569 (19.2)	2910 (17.5)	5003 (17.9)	1154 (10.5)	3666 (14.8)	
Unknown	1147 (0.5)	827 (0.5)	54 (0.3)	79 (0.3)	90 (0.8)	97 (0.4)	
**Levels of physical activities, n (%)**							<0.001
Low	76,618 (32.0)	47,554 (29.9)	5964 (35.9)	9211 (33.0)	4867 (44.1)	9022 (36.4)	
Moderate	85,403 (35.7)	57,868 (36.4)	5758 (34.6)	9893 (35.4)	3341 (30.3)	8543 (34.5)	
High	77,415 (32.3)	53,661 (33.7)	4905 (29.5)	8816 (31.6)	2833 (25.7)	7200 (29.1)	
**Alcohol consumption, n (%)**							<0.001
Never	22,751 (9.5)	12,842 (8.1)	1952 (11.7)	2650 (9.5)	2201 (19.9)	3106 (12.5)	
Once or twice a week or less	128,606 (53.7)	84,178 (52.9)	8816 (53.0)	14,979 (53.6)	6553 (59.4)	14,080 (56.9)	
Three times a week or more	87,417 (36.5)	61,568 (38.7)	5819 (35.0)	10,247 (36.7)	2255 (20.4)	7528 (30.4)	
Unknown	662 (0.3)	495 (0.3)	40 (0.2)	44 (0.2)	32 (0.3)	51 (0.2)	
**Ethnicity, n (%)**							<0.001
White	224,792 (93.9)	149,010 (93.7)	15,960 (96.0)	26,260 (94.1)	9802 (88.8)	23,760 (95.9)	
Asia	5200 (2.2)	3615 (2.3)	192 (1.2)	558 (2.0)	508 (4.6)	327 (1.3)	
Black and Caribbean	4286 (1.8)	2975 (1.9)	146 (0.9)	491 (1.8)	427 (3.9)	247 (1.0)	
Other/unknown	5158 (2.2)	3483 (2.2)	329 (2.0)	611 (2.2)	304 (2.8)	431 (1.7)	
**Region, n (%)**							<0.001
England	212,190 (88.6)	140,684 (88.4)	15,006 (90.3)	24,840 (89.0)	9744 (88.3)	21,916 (88.5)	
Scotland	17,382 (7.3)	11,914 (7.5)	1022 (6.1)	1786 (6.4)	837 (7.6)	1823 (7.4)	
Wales	9864 (4.1)	6485 (4.1)	599 (3.6)	1294 (4.6)	460 (4.2)	1026 (4.1)	
**Socioeconomic status based on Townsend** **Score, n (%)**							<0.001
Interquartile 1	59,168 (24.7)	40,773 (25.6)	3653 (22.0)	6715 (24.1)	1904 (17.2)	6123 (24.7)	
Interquartile 2	58,909 (24.6)	40,010 (25.2)	3918 (23.6)	6477 (23.2)	2333 (21.1)	6171 (24.9)	
Interquartile 3	59,853 (25.0)	39,856 (25.1)	4195 (25.2)	6949 (24.9)	2708 (24.5)	6145 (24.8)	
Interquartile 4	61,506 (25.7)	38,444 (24.2)	4861 (29.2)	7779 (27.9)	4096 (37.1)	6326 (25.5)	

IQR: Interquartile range. * *p*-value expresses the presence of statistically significant differences among the five morbidity patterns identified (Kruskal–Wallis test for continuous variables, Pearson’s χ^2^ test for categorical). ^µ^ Post-menopausal women only.

**Table 2 cancers-15-01165-t002:** Association between preexisting single diseases at baseline and breast cancer risk.

Pre-Existing Disease at Baseline	Number of Breast Cancer Cases/Person Years	Age-Adjusted Model HR (95%CI)	Multivariable Model HR (95%CI)
**Hypertension**			
No	3979/1,287,967	1.00 (Reference)	1.00 (Reference)
Yes	1347/383,417	1.06 (0.99–1.13)	1.03 (0.97–1.11)
**Pain condition**			
No	4336/1,386,565	1.00 (Reference)	1.00 (Reference)
Yes	990/284,820	1.06 (0.98–1.13)	1.04 (0.97–1.12)
**Asthma**			
No	4692/1,465,134	1.00 (Reference)	1.00 (Reference)
Yes	634/206,250	0.97 (0.89–1.05)	0.96 (0.88–1.04)
**Thyroid disorders**			
No	4836/1,513,768	1.00 (Reference)	1.00 (Reference)
Yes	490/157,617	0.94 (0.85–1.03)	0.93 (0.85–1.02)
**Treated dyspepsia**			
No	4898/1,548,551	1.00 (Reference)	1.00 (Reference)
Yes	428/122,834	1.04 (0.95–1.15)	1.04 (0.94–1.15)
**Depression**			
No	4927/1,557,562	1.00 (Reference)	1.00 (Reference)
Yes	399/113,821	1.13 (1.02–1.26)	1.12 (1.01–1.24)
**Migraine**			
No	5099/1,601,276	1.00 (Reference)	1.00 (Reference)
Yes	227/70,109	1.04 (0.91–1.18)	1.05 (0.91–1.19)
**Psoriasis**			
No	5131/1,612,546	1.00 (Reference)	1.00 (Reference)
Yes	195/58,839	1.06 (0.92–1.22)	1.04 (0.90–1.2)
**Diabetes**			
No	5138/1,616,001	1.00 (Reference)	1.00 (Reference)
Yes	188/55,384	1.02 (0.88–1.18)	0.99 (0.85–1.15)
**Irritable bowel syndrome**			
No	5157/1,617,608	1.00 (Reference)	1.00 (Reference)
Yes	169/53,776	0.98 (0.84–1.15)	0.99 (0.85–1.15)
**Rheumatoid arthritis**			
No	5181/1,624,015	1.00 (Reference)	1.00 (Reference)
Yes	145/473,698	0.92 (0.78–1.09)	0.92 (0.78–1.09)
**Coronary heart disease**			
No	5227/1,632,796	1.00 (Reference)	1.00 (Reference)
Yes	99/38,589	0.72 (0.59–0.88)	0.73 (0.60–0.89)
**Anxiety**			
No	5221/1,637,202	1.00 (Reference)	1.00 (Reference)
Yes	105/34,183	0.97 (0.80–1.18)	0.96 (0.79–1.17)
**COPD**			
No	5245/1,648,455	1.00 (Reference)	1.00 (Reference)
Yes	81/22,930	1.05 (0.84–1.30)	1.07 (0.86–1.33)
**Stroke**			
No	5260/1,649,817	1.00 (Reference)	1.00 (Reference)
Yes	66/21,568	0.89 (0.70–1.14)	0.91 (0.71–1.16)
**Diverticular disease of intestine**			
No	5258/1,650,114	1.00 (Reference)	1.00 (Reference)
Yes	68/21,271	0.92 (0.72–1.17)	0.9 (0.71–1.15)

HR: hazard ratio; CI: confidence interval; the fully adjusted model was adjusted for age at baseline (continuous), age at menarche (continuous), age at menopause (still had periods; had menopause before the age of 45 years; had menopause between the age of 45 and 54; had menopause after the age of 55), menopausal hormone therapy use (never; ever, less than 5-year duration; ever, 5 years and longer; ever, unknown duration), oral contraceptive use (never; ever, less than 10-year duration; ever, at least 10-year duration; ever, unknown duration; unknown status), parity and age at first birth (no live birth; at least one birth before age 30; at least one birth after age 30), BMI (continuous), ethnicity (Asian; Black/Caribbean; White; others/unknown), Townsend score (continuous); level of physical activity (low; moderate; high), alcohol consumption (never; twice a week or less; three times a week or more; unknown status).

**Table 3 cancers-15-01165-t003:** Associations among number of morbidities, morbidity patterns, and breast cancer risk.

	Study Population (n = 239,436)			Postmenopausal Women Only (n = 175,949)		
Characteristics	Breast Cancer Cases/Person- Years	Age-Adjusted Models HR (95%CI)	Fully Adjusted Models HR (95%CI)	Breast Cancer Cases/Person Years	Age-Adjusted Models HR (95%CI)	Fully Adjusted Models HR (95%CI)
**Number of morbidities**						
No morbidity	2131/69,8776	1.00 (Reference)	1.00 (Reference)	1451/454,566	1.00 (Reference)	1.00 (Reference)
One morbidity	1736/54,3974	1.01 (0.95–1.08)	1.00 (0.94–1.07)	1361/408,943	1.02 (0.95–1.10)	1 (0.93–1.08)
Multi-morbidities	1459/428,635	1.04 (0.97–1.02)	1.03 (0.96–1.11)	1268/359,844	1.06 (0.98–1.14)	1.02 (0.94–1.1)
Two morbidities	911/266,831	1.05 (0.97–1.14)	1.04 (0.96–1.13)	786/218,780	1.08 (0.99–1.18)	1.04 (0.95–1.14)
3+ morbidities	548/161,804	1.03 (0.93–1.13)	1.01 (0.92–1.12)	482/141,065	1.02 (092–1.14)	0.97 (0.87–1.08)
**Morbidity patterns**						
No-predominant morbidity	3534/1,110,979	1.00 (Reference)	1.00 (Reference)	2670/798,572	1.00 (Reference)	1.00 (Reference)
Psychiatric morbidities	381/115,476	1.06 (0.95–1.18)	1.04 (0.94–1.16)	264/80,575	1.00 (0.88–1.14)	0.98 (0.86–1.11)
Respiratory/immunological morbidities	611/195,129	0.99 (0.91–1.08)	0.98 (0.9–1.07)	467/137,526	1.02 (0.92–1.12)	1.01 (0.91–1.11)
Cardiovascular/metabolic morbidities	246/75,843	0.94 (0.83–1.07)	0.93 (0.81–1.06)	232/69,252	0.96 (0.84–1.10)	0.91 (0.79–1.05)
Unspecific morbidities	554/173,957	0.98 (0.89–1.07)	0.98 (0.89–1.07)	447/137,429	0.96 (0.87–1.06)	0.95 (0.86–1.05)

HR: hazard ratio; CI: confidence interval; the fully adjusted model was adjusted for age at baseline (continuous), age at menarche (continuous), age at menopause (still had periods; had menopause before the age of 45 years; had menopause between the age of 45 and 54; had menopause after the age of 55; others/unknown), Townsend score (continuous); level of physical activity (low; moderate; high), alcohol consumption (never; once or twice a week or less; three times a week or more; unknown status), menopausal hormone therapy use (never; ever, less than 5-year duration; ever, 5 years and longer; ever, unknown duration), oral contraceptive use (never; ever, less than 10-year duration; ever, at least 10-year duration; ever, unknown duration; unknown status), parity and age at first birth (no live birth; at least one birth before age 30; at least one birth after age 30), BMI (continuous), ethnicity (Asian; Black/Caribbean; White).

**Table 4 cancers-15-01165-t004:** Association between morbidity patterns and breast cancer risk, counting death and first diagnosed non-breast cancer cases as a competing risk.

Event	Morbidity Pattern	Cases/Person-Years	Hazard Ratio (95%CI)
**Breast cancer as first diagnosed cancer**			
	No-predominant morbidity	3534/1,110,979	1.00 (Reference)
	Psychiatric morbidities	381/115,476	1.04 (0.94–1.16)
	Respiratory/immunological morbidities	611/195,129	0.98 (0.90–1.07)
	Cardiovascular/metabolic morbidities	246/758,423	0.93 (0.81–1.06)
	Unspecific morbidities	554/173,957	0.98 (0.89–1.07)
**Non-breast cancer as first diagnosed cancer**			
	No-predominant morbidity	4964/1,110,979	1.00 (Reference)
	Psychiatric morbidities	485/115,476	0.96 (0.88–1.06)
	Respiratory/immunological morbidities	1041/195,129	1.18 (1.11–1.27)
	Cardiovascular/metabolic morbidities	561/758,423	1.19 (1.09–1.30)
	Unspecific morbidities	862/173,957	1.00 (0.93–1.07)
**Death**			
	No-predominant morbidity	645/1,110,979	1.00 (Reference)
	Psychiatric morbidities	126/115,476	1.82 (1.50–2.21)
	Respiratory/immunological morbidities	203/195,129	1.68 (1.44–1.97)
	Cardiovascular/metabolic morbidities	242/758,423	3.06 (2.61–3.58)
	Unspecific morbidities	205/173,957	1.65 (1.41–1.94)

HR: hazard ratio; CI: confidence interval. The model was adjusted for age at baseline (continuous), age at menarche (continuous), age at menopause (still had periods; had menopause before the age of 45 years; had menopause between the age of 45 and 54; had menopause after the age of 55), menopausal hormone therapy use (never; ever, less than 5-year duration; ever, 5 years and longer; ever, unknown duration), oral contraceptive use (never; ever, less than 10-year duration; ever, at least 10-year duration; ever, unknown duration; unknown status), parity and age at first birth (no live birth; at least one birth before age 30; at least one birth after age 30), BMI (continuous), ethnicity (Asian; Black/Caribbean; White; others/unknown), Townsend score (continuous); level of physical activity (low; moderate; high), alcohol consumption (never; twice a week or less; three times a week or more; unknown status).

## Data Availability

This work has been conducted using the UK Biobank Resource under Application Number 35032. Bona-fide researchers can apply to use the UK Biobank dataset by registering and applying at http://www.ukbiobank.ac.uk/register-apply accessed on 1 March 2022.

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
