# Peer review of "Multi-Morbidity and Risk of Breast Cancer among Women in the UK Biobank Cohort"

_cancers, 2023, doi:10.3390/cancers15041165_

Round 1

Reviewer 1 Report

The manuscript is well written and presented. few minor comments must be addressed before publications:

Table 1. The figure captions should define the number between brackets as SE or SD.

Also, the authors should minimize the use of pronoun we and rewrite the sentence using the passive voice.

Author Response

 Thank you for your time and constructive comments on our manuscript. Below are our responses to your comments.

  • Table 1: The numbers between brackets are either percentage or interquartile range and we have - specified these detailed in the first column of the table
  • Most journal editors prefer the active voice over the passive voice since the former emphasizes author responsibility, improves readability, and reduces ambiguity. In our manuscript, we aim to deliver our message in a direct, clear, and concise way. We will defer to the Editor and the editorial policies of the journal regarding this point.

Reviewer 2 Report

Dear  Editor-In-Chief

This is an innovative work to determine the risk factors of the most prevalent cancer among women worldwide.

interesting findings are not a relation between chronic disease to breast cancer and the relation between psychiatric patterns and chronic stress to induce it. which is confirmed in all research as well.

Author Response

Thank you for your time working on our manuscript

Reviewer 3 Report

Dear authors,

I read with interest the article which analyzes the importance of to investigate the association between

morbidity patterns and breast cancer risk.

It is also interesting to further analyze the pathogenic traits of mental disorders for further genetic insights, since these parameters are important for the progressive improvement of therapeutic pathways.

I thank the authors for carefully addressing this issue.

I recommend this article.

The manuscript is quite clear, relevant for the field and presented in a quite well-structured manner.

The cited references mostly recent publications (within the last 5 years) are relevant.

The manuscript scientifically sound and is the experimental design appropriate to test the hypothesis.

The manuscript’s results reproducible based on the details given in the methods section..

The figures tables are appropriate and they properly show the data (easy to interpret and understand , to be implemented). 

In conclusion the question original and well-defined and the results provide an advancement of the current knowledge. In this regard the work fit the journal scope.

The conclusions interesting for the readership of the journal and will the paper attract a wide readership.

Author Response

(The authors gave the same response as above.)
